# Laminar flow ventilation system to prevent airborne infection during exercise in the COVID-19 crisis: A single-center observational study

Yoshinori Katsumata[1,2]*, Motoaki Sano[2], Hiroki Okawara[3], Tomonori Sawada[3], Daisuke Nakashima[3], Genki Ichihara[2], Keiichi Fukuda[2], Kazuki Sato[1], Eiji Kobayashi[4]

1 Institute for Integrated Sports Medicine, Keio University School of Medicine, Tokyo, Japan, 2 Department of Cardiology, Keio University School of Medicine, Tokyo, Japan, 3 Department of Orthopaedic Surgery, Keio University School of Medicine, Tokyo, Japan, 4 Department of Organ Fabrication, Keio University School of Medicine, Tokyo, Japan

* goodcentury21@keio.jp

## Abstract

Particulate generation occurs during exercise-induced exhalation, and research on this topic is scarce. Moreover, infection-control measures are inadequately implemented to avoid particulate generation. A laminar airflow ventilation system (LFVS) was developed to remove respiratory droplets released during treadmill exercise. This study aimed to investigate the relationship between the number of aerosols during training on a treadmill and exercise intensity and to elucidate the effect of the LFVS on aerosol removal during anaerobic exercise. In this single-center observational study, the exercise tests were performed on a treadmill at Running Science Lab in Japan on 20 healthy subjects (age: 29±12 years, men: 80%). The subjects had a broad spectrum of aerobic capacities and fitness levels, including athletes, and had no comorbidities. All of them received no medication. The exercise intensity was increased by 1-km/h increments until the heart rate reached 85% of the expected maximum rate and then maintained for 10 min. The first 10 subjects were analyzed to examine whether exercise increased the concentration of airborne particulates in the exhaled air. For the remaining 10 subjects, the LFVS was activated during constant-load exercise to compare the number of respiratory droplets before and after LFVS use. During exercise, a steady amount of particulates before the lactate threshold (LT) was followed by a significant and gradual increase in respiratory droplets after the LT, particularly during anaerobic exercise. Furthermore, respiratory droplets ≥0.3 μm significantly decreased after using LFVS (2120800±759700 vs. 560 ± 170, $p<0.001$). The amount of respiratory droplets significantly increased after LT. The LFVS enabled a significant decrease in respiratory droplets during anaerobic exercise in healthy subjects. This study's findings will aid in exercising safely during this pandemic.

**Data Availability Statement:** All relevant data are within the manuscript and its Supporting information files.

**Funding:** Nippon Medical & Chemical Instruments Co. Ltd. developed the laminar airflow ventilation system and provided support in the form of financial supports for authors YK and EK. This work was also partly supported by a Grant-in-Aid from Scientific Research from the Japan Agency for Medical Research and Development [ID. JP21ek0210130] and by Kimura Memorial Heart Foundation Research Grant for 2019 [N/A], Suzuken Memorial Foundation [N/A], Foundation for Total Health Promotion [N/A], and Research Grant for Public Health Science[N/A]. The did not have any additional role in the study design, data collection and analysis, decision to publish, or preparation of the manuscript. The specific roles of these authors are articulated in the 'author contributions' section.

**Competing interests:** The authors have read the journal's policy and have the following competing interests: YK has a financial relationship with Kimura Memorial Heart Foundation Research Grant for 2019, Suzuken Memorial Foundation, Foundation for Total Health Promotion, and Research Grant for Public Health Science. EK and YK have a financial relationship with the research grant from Nippon Medical & Chemical Instruments Co. Ltd. DN is the shareholder and CEO of Grace imaging Inc. which provided the lactate sensor equipment. MS, HO, TS, GI, and KS declare that they have no competing interests. This does not alter our adherence to PLOS ONE policies on sharing data and materials.

**Abbreviations:** COVID-19, coronavirus disease; HEPA, high-efficiency particulate absorbing; LFVS, laminar airflow ventilation system; LT, lactate threshold; SARS-CoV-2, severe acute respiratory syndrome coronavirus 2; $VCO_2$, increase in $CO_2$ consumption; VE, minute ventilation; $VO_2$, oxygen uptake.

## Introduction

Human-to-human transmission of the severe acute respiratory syndrome coronavirus 2 (SARS-CoV-2) mainly occurs through respiratory droplets emitted by asymptomatic infected individuals [1, 2]. After the end of the state of emergency and reopening of gyms, many individuals no longer felt comfortable going to the gym [3, 4]. In poorly ventilated and enclosed spaces, exercise-induced labored breathing can result in higher concentrations of virus-containing respiratory droplets from an infected person, thereby increasing the risk of airborne transmission if another individual shares or enters the same space soon after an infected person leaves [5–7]. However, literature on the extent of particulate generation during exercise-induced exhalation is scarce [8], and scientific infection-control measures are inadequately implemented. Therefore, it is urgent to apply an innovative system to prevent SARS-CoV-2 infection during exercise, especially indoor fitness activities.

The laminar airflow ventilation system (LFVS) combined with high-efficiency particulate absorbing (HEPA) filters is used in operating rooms and clean benches to remove moderate-to-large-sized fractions of aerosols from airstreams inside rooms [9, 10]. HEPA filters can remove at least 99.95% of particulates with diameters >0.3 μm. Therefore, the LFVS technology was used to develop a preventive measure against airborne infection among treadmill users during the coronavirus disease (COVID-19) crisis. This study aimed to investigate the relationship between the number of aerosols during training on a treadmill and exercise intensity and to elucidate the effectiveness of the LFVS in aerosol removal during anaerobic exercise.

## Materials and methods

### LFVS

A considerable amount of airflow would be required to make the entire room sterile. Therefore, the LFVS was developed as a booth to remove only the respiratory droplets in the exhalation area of runners (Figs 1 and 2). The booth has a frame structure with a height of 2140 mm, width of 1000 mm, and depth of 1560 mm, and it is separated from the surrounding area using a vinyl sheet. Air was emitted from the ceiling-mounted air supply unit above the treadmill (push) and drawn in the exhaust unit on the floor in front of the treadmill (pull; Figs 1 and 2) to maintain a constant vertical laminar flow from the ceiling to the treadmill's running plate. Airflow and exhaust volume were optimized by verifying changes in the airflow from the humidifier. The airflow rate of the air supply unit is 12 m³/min, creating a laminar flow from the air-blowing surface to the exhaled area of the runner at a velocity of approximately 0.5 m/s. The exhaust unit has a treatment air volume of 16 m³/min. The treatment air volume of the exhaust unit is 16 m³/min. Thus, the LFVS facilitated vertical displacement of aerosolized droplets of various sizes sprayed horizontally from the runner's mouth and nose during respiration. Moreover, HEPA filters were installed on both units to trap the virus (Fig 1).

### Number of dust particulates sprayed from the ultrasonic humidifier

The performance of the LFVS was evaluated on a stand-alone unit without a treadmill in the absence of frontal disturbance, using artificial droplets. An ultrasonic humidifier was used to spray 1 L/h of steam from the discharge port (30 mm diameter) at a position assumed to be the breathing area, and the number of dust particulates was measured by a particulate counter (RION, Tokyo, Japan) at a distance 20 cm away from the blowout port. The particulate counter measured the number of particulates per cf (28.3 L) through a 1/100 diluter (TOPAS GmbH, Dresden Germany). Particulates (>0.3 μm) were measured every 1 minute, 10 times in an

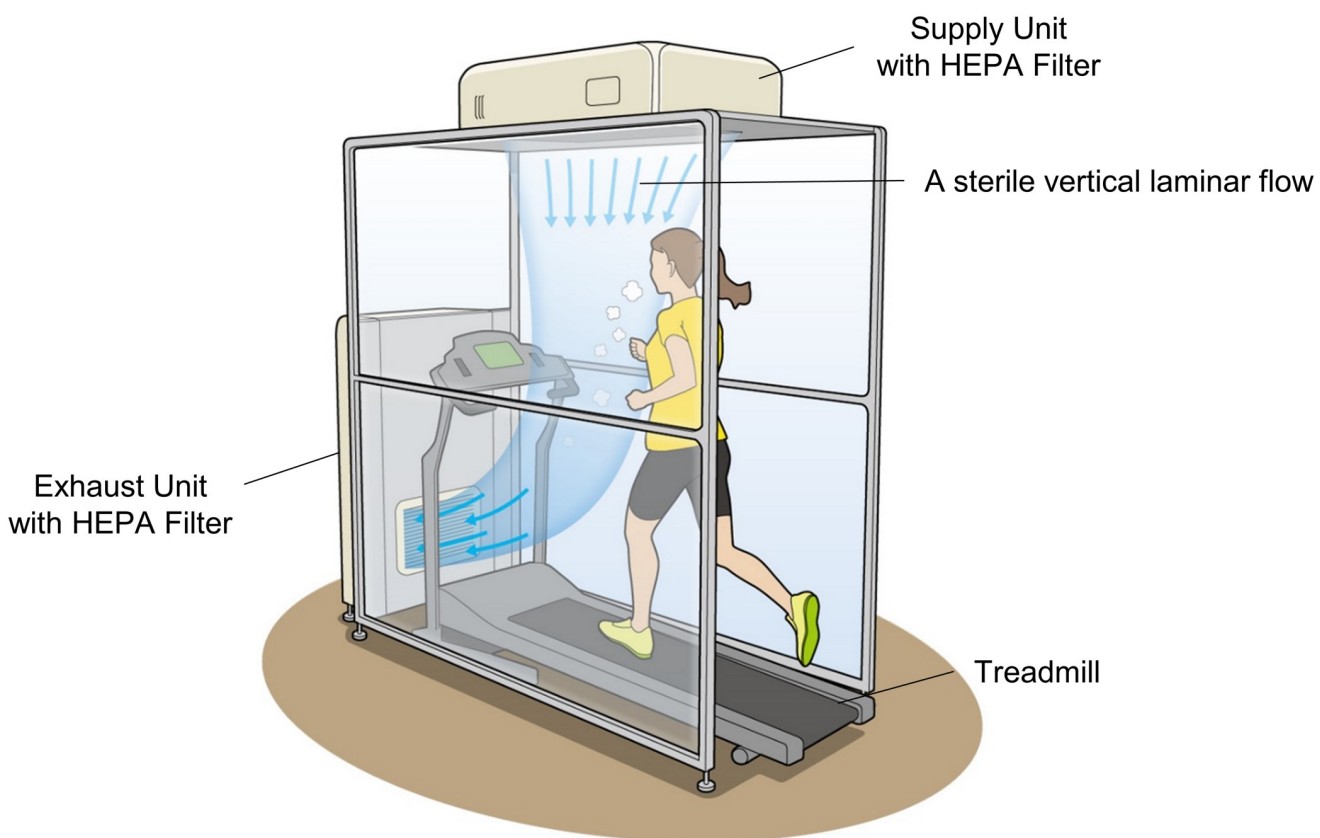

**Fig 1. Laminar airflow ventilation system comprising an air supply unit and an exhaust unit.** The air supply unit ensures high-efficiency particulate air (HEPA)-filtered vertical airflow from over the runner's head, whereas the exhaust unit draws contaminated air into the treadmill board. A booth encloses all three sides except for the entrance to the treadmill to maintain the laminar flow. The air sanitized by the HEPA filter in the exhaust unit is vented outside the booth.

unloaded state with a humidifier, a loaded state with a humidifier, and a loaded state with a humidifier and the LFVS activated.

Air was emitted from the ceiling-mounted air supply unit above the treadmill and drawn in from the exhaust unit on the floor. Airflow and exhaust volume were optimized by visually verifying changes in the humidifier's airflow. By constructing air filtration using the HEPA filter in the exhaust unit, this system enabled the capture of respiratory droplets containing SARS--CoV-2 emitted by asymptomatic infected individuals.

## The airflow that formed in the booth

The airflow was visualized using an airflow indicator tube (no. 301, Komyo Rikagaku Kogyo K.K. Kanagawa, Japan) located at a position assumed to be the breathing area (Fig 2A and 2B and S1 Movie). The smoke diffusion pattern was captured in different states, including a non-activated LFVS (OFF), only the exhaust unit activated (Exhaust ON), only the air supply unit activated (Supply ON), and the LFVS activated (Supply and Exhaust ON). The smoke was stagnant owing to the updraft in an unloaded state with the airflow indicator tube (Fig 2C). Smoke was drawn in the exhaust unit while lingering inside the booth when activating only the exhaust unit (Fig 2D). Smoke diffusion was controlled, but some smoke leaked outside the booth, activating only the air supply unit (Fig 2E).

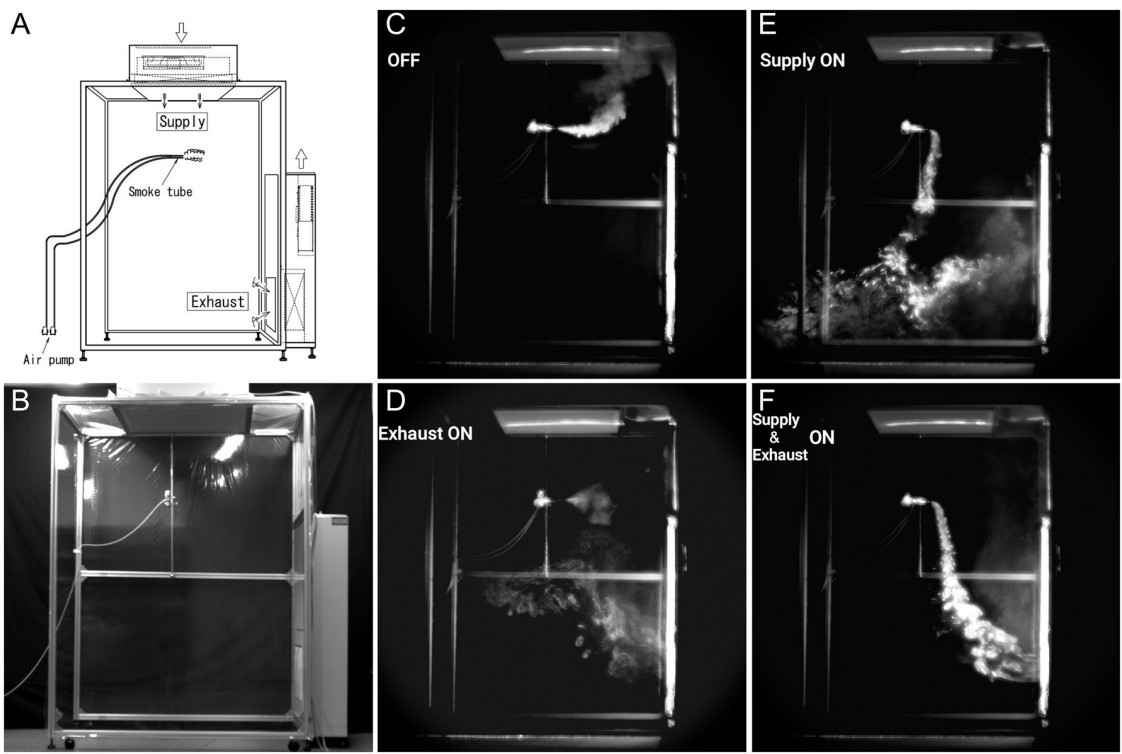

**Fig 2. Performance of the laminar airflow ventilation system (LFVS) using artificial droplets.** (A and B) The airflow was observed using an airflow indicator tube located at a position assumed to be the breathing. (C–F) The four types of vertical airflow, including (C) not-activating the LFVS, D: activating only the exhaust unit (Exhaust ON), (E) activating only the air supply unit (Supply ON), (F) activating the two units (Supply and Exhaust ON).

## Study sample and ethical approval

Subjects aged 20–80 years were recruited via a web system in October 2020. Exclusion criteria included receiving medication and having comorbidities, such as hypertension, diabetes, or active lung diseases. Twenty healthy subjects enrolled in this study had a broad spectrum of aerobic capacities and fitness levels, including athletes. One of them had experienced a subdural hematoma 6 months ago. At the time he participated in this study, he was cured of his disease and was not receiving ongoing treatment. In addition, he was actively engaged in daily exercise. All of them received no medication. They can be considered representative of a larger population. The study protocol was approved by the Institutional Review Board (IRB) of Keio University School of Medicine [permission number; 20190229], and the study was conducted in accordance with the Declaration of Helsinki. Subjects provided verbal informed consent because the IRB approved use of oral consent in accordance with Japanese guidance for clinical research. Verbal consents were recorded as experimental notes in this study.

## Experimental procedure

The exercise tests were performed on a treadmill at Running Science Lab in Japan, simultaneously monitoring particulates emitted by subjects with a particulate counter every minute. During exercise, the lactate concentration in sweat was monitored with a sweat lactate sensor attached to the upper arm (Graceimaging Inc., Tokyo, Japan) to determine the lactate

threshold (LT), [11] and the heart rate was monitored using Duranta (Zaiken, Tokyo, Japan). The first 10 subjects were registered to examine whether exercise increases airborne particulate concentration in the exhaled air. The other 10 subjects were registered to investigate the effective removal of micro-droplets emitted during exhalation when running on the treadmill using the LFVS. The two groups exercised according to the same protocol, and the LFVS was activated during constant exercise for 4 minutes after reaching 85% of the expected maximum heart rate. Then, the exercise was continued for another 6 minutes, activating the LFVS.

## Exercise testing protocol

On the day of the exercise test, the subjects avoided heavy physical activity before the test. The subjects performed the test in the upright position on a treadmill (Elevation Series®, Life Fitness, Illinois, USA). The subjects performed a 5-minute warm-up from 5 to 10 km/h with a 1-degree incline according to their conditions after a 2-minute rest to stabilize the heart rate and respiratory condition. Then, the exercise intensity was gradually increased by 1-km/h increments, followed by 10 minutes of constant-load exercise at 85% of the expected maximum heart rate (220 − age). For some subjects, the loading volume was fine-tuned to maintain the heart rate at 85%.

## Particulates count in exercise

A particulate counter placed 10 cm away from the treadmill runner's mouth measured the exhaled air's droplet concentration. After being diluted 100-fold in a diluter, the number of particulates was analyzed using an instrument. The number of particulates was measured every minute from 2 minutes before starting the exercise through to the end.

## LT in sweat

LT was defined as the first significant increase in the lactate concentration in sweat above the baseline based on the graphical plots [11]. Three researchers, independent of the researchers who analyzed respiratory gas exchange, jointly agreed on the point of LT.

## Statistical analyses

The results are presented as means with standard deviations for continuous variables and as percentages for categorical variables, as appropriate. Based on a pre-performed Shapiro-wilk test, multiple comparisons of changes (Δ) in the number of airborne particulates involving each incremental exercise period from the warm-up were made using the repeated analysis of variance with the Dunnet test as a post-hoc test. Student's paired *t*-test was used to compare the droplet concentrations from spray or from the oral cavity during vigorous exercise before and after the activation of LFVS. Cohen's d was calculated using the value of t for paired t test. SPSS, version 25.0 (SPSS Inc., Chicago, Illinois), was used for analysis, and $p < 0.05$ (2-sided) was set to define statistical significance.

## Patient and public involvement

There was no active patient involvement in the design of the study or in the recruitment to, or conduct of, the study. The subjects participated in this study after receiving an explanation of the protocol approved by the Institutional Review Board of the Keio University School of Medicine.

**Table 1. Baseline characteristics of patients.**

| Characteristics | | Total | Off | On | *P*-value |
|---|---|---|---|---|---|
| **Age, years** | | 29 ± 12 | 24 ± 6 | 34 ± 14 | 0.08 |
| **Male, n (%)** | | 16 (80) | 9 (90) | 7 (70) | |
| **Height, cm** | | 169 ± 8 | 171 ± 5 | 167 ± 9 | 0.30 |
| **Body weight, kg** | | 60 ± 12 | 64 ± 13 | 56 ± 9 | 0.11 |
| **BMI, kg/m²** | | 20.8 ± 2.6 | 21.8 ± 3.1 | 19.8 ± 1.7 | 0.10 |
| **Exercise** | **>3 times/week, n (%)** | 10 (50) | 4 (40) | 6 (60) | |
| | **1–2 times/week, n (%)** | 6 (30) | 3 (30) | 3 (30) | |
| | **Sedentary life, n (%)** | 4 (20) | 3 (30) | 1 (10) | |
| **Load at 85% of the EMHR, km/h** | | 13 (3) | 13 (3) | 14 (4) | 0.38 |

**Abbreviations**: BMI; body mass index, EMHR; expected maximum heart rate.

## Results

### Non-clinical study of the LFVS

A constant vertical laminar flow was observed only when the LFVS was activated but not when the air supply unit or exhaust unit was activated (Fig 2F and S1 Movie). S1 Fig shows the response of the LFVS to emitting particulates from the ultrasonic humidifier. The particulates of >0.3 μm sprayed from ultrasonic humidifier were almost completely removed by the LFVS (3717040 ± 72347 vs. 60 ± 84; $p < .001$, n = 10) as well as particulates of sizes >0.5 μm (327780 ± 22908 vs. 0±0; $p<0.001$, n = 10) and >1.0 μm (23620 ± 2504 vs. 0±0; $p<0.001$, n = 10).

### Monitoring particulates during exercise

Baseline characteristics of the subjects are summarized in Table 1. The subjects were predominantly male (80%), with an average age of 29 ± 12 years. Fig 3 shows the representative data for the particulates, lactate in sweat, and heart rate during exercise. A steady number of particulates before the LT was followed by a significant and gradual increase in the respiratory droplets after the LT, particularly during anaerobic exercise with a large effect size (Fig 3 and Table 2).

### Effect of the LFVS on the particulates during vigorous exercise

The LFVS was activated during a constant load of exercise intensity above the LT, in which the concentration of particulates in the exhaled air increased. Notably, particulates exhaled during exercise were almost completely removed by the LFVS (>0.3 μm: 2120800±759700 vs. 560 ± 170; $p<0.001$, $p<0.001$, n = 10; Fig 4 and Table 3).

## Discussion

The most striking result among our findings is that exercise below the LT did not increase the particulate concentration in the exhaled air. In contrast, higher-intensity anaerobic exercise continuation increased the exhaled droplet concentration. In addition, the LFVS enabled a significant decrease in particulates emitted from the oral cavity during vigorous exercise.

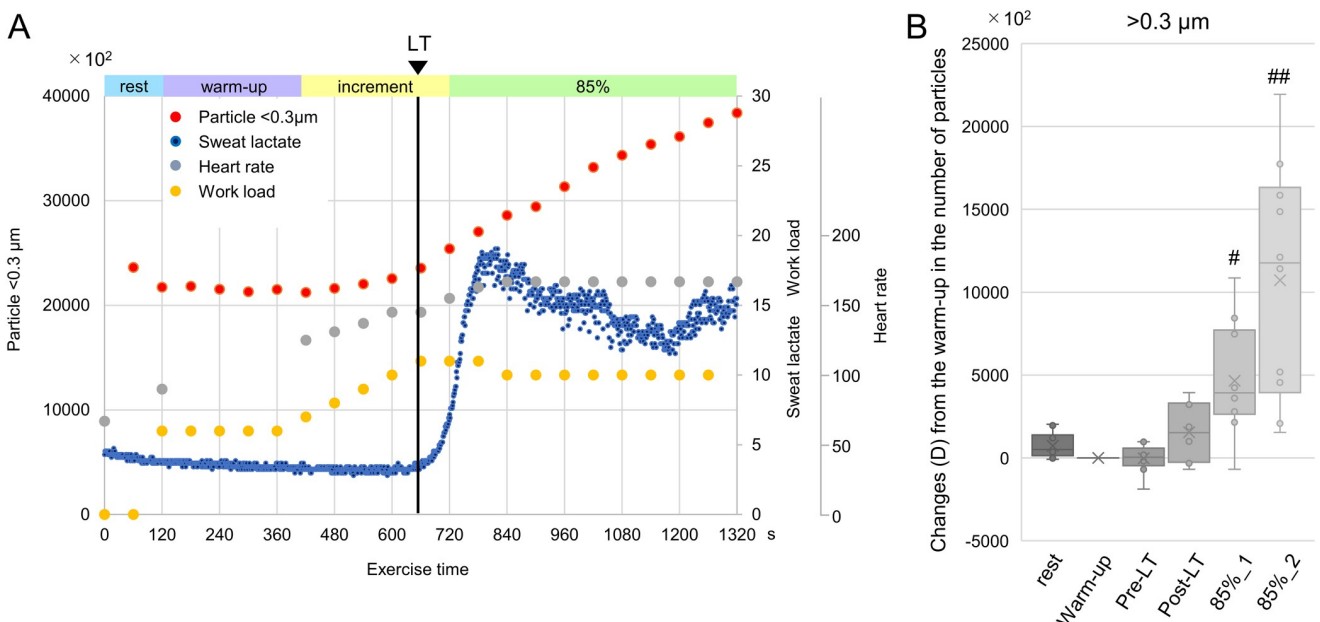

**Fig 3. Airborne particulates generation during exercise.** (A) Representative graphs of the concentration of airborne particulates (<0.3 μm; red dots), sweat lactate (blue dots) levels, and heart rate (gray dots) during exercise. (B) Changes (Δ) from the warm-up in the number of airborne particulates (>0.3 μm; n = 10). [#]$p<0.01$, [##]$p<0.001$ compared with the warm-up. LT, lactate threshold; 85%_1, the first half of the constant-load exercise at 85% of the expected maximum heart rate; 85%_2, the latter half of the constant-load exercise at 85% of the expected maximum heart rate.

## LFVS during treadmill exercise

Adequate regular physical activity is paramount to maintaining good health [12–16]. Moreover, current clinical practice guidelines and expert statements recommend that adults should engage in at least 150 minutes of moderate-intensity aerobic physical activity, which consists of at least 75 minutes of vigorous-intensity aerobic physical exercise each week [17]. However, indoor exercise, such as in a training gym, had been banned during the state of emergency due to the COVID-19 pandemic in many countries [3]. Furthermore, decreased physical activity and associated worse depression, loneliness, stress have been observed [18–21]. The virus can spread by close contact and indirect contact through contaminated objects [2, 22]. Indoor sports centers have been re-opened after the end of the state of emergency, and the infection has been controlled with hand washing, social distancing, and prohibiting high-intensity

**Table 2. Airborne particulate generation during exercise.**

|  | rest | Warm-up | Pre-LT | Post-LT | 85%_1 | 85%_2 |
|---|---|---|---|---|---|---|
| Mean difference ± SD (95%CI) | 731.3 ± 760.0 (-2727.6–4190.2) | - | -27.5 ± 850.9 (-3591.9–3537.0) | 1552.4 ± 1667.4 (-2012.0–5116.9) | 4653.4 ± 3380.5 (1088.9–8217.8) | 10727.3 ± 7060.6 (7162.8 ± 14291.7) |
| p-value[#] | 0.97 | - | 1.00 | 0.64 | < 0.01 | < 0.001 |
| Power | 0.77 | - | 0.05 | 0.75 | 1.00 | 1.00 |

Repeated analysis of variance with Dunnet test as post-hoc test revealed a significant and gradual increase in the respiratory droplets with time.

[#]$p$ value compared with the warm-up.

LT, lactate threshold; 85%_1, the first half of the constant-load exercise at 85% of the expected maximum heart rate; 85%_2, the latter half of the constant-load exercise at 85% of the expected maximum heart rate.

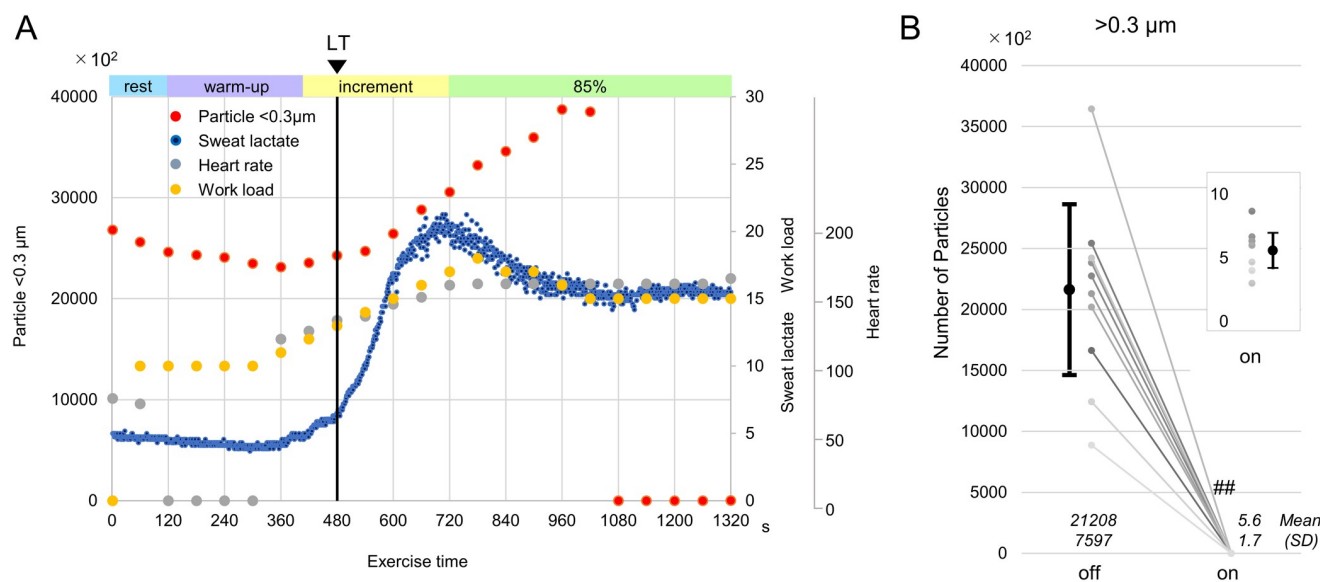

**Fig 4. Airborne particulate generation during anaerobic exercise using the laminar airflow ventilation system (LFVS).** (A) Representative graphs of the concentration of airborne particulates (<0.3 μm) (red dots) in activating the LFVS during constant exercise 4 minutes after reaching 85% of the expected maximum heart rate. (B) Concentration of airborne particulates (>0.3 μm) before and after the activation of the LFVS (n = 10). ##$p$<0.001 compared with the off LFVS, 85%, constant-load exercise at 85% of the expected maximum heart rate, LT, lactate threshold.

exercise [23]. Of respiratory droplets during exercise, droplets with a diameter of 10 μm or more fall to the ground immediately after traveling a distance of 1–2 m. However, smaller droplets (i.e., aerosols) are known to remain suspended in the air for hours [24, 25]. Recently, the virus has been found in small aerosols and has been shown to maintain viability in such aerosols for hours [5, 26]. Thus, it would be judicious to take precautionary measures to control airborne transmission of infection. Particularly, when using treadmill training installed in many gyms, preventing infection from droplets by simply securing social distancing or space partitioning is challenging. The use of efficient ventilation, high-efficiency air filters, and ultraviolet lamps is expected to overcome this problem. Based on the guidelines for the prevention of surgical site infection issued by the Centers for Disease Control and Prevention [9], a ventilation system (LFVS) to create a sterile vertical laminar flow was developed. A considerable amount of airflow would be required to make the entire room sterile. Therefore, the LFVS was developed as a booth to remove the respiratory droplets only in the exhalation area of runners (Fig 1). It was shown that pull alone through the exhaust or push alone from the supply unit was not effective in removing particulates. However, the LFVS effectively and completely prevented the diffusion of particulates into the room, even under high-intensity training

**Table 3. Airborne particulate generation during anaerobic exercise using the laminar airflow ventilation system (LFVS).**

|  | LFVS off | LFVS on | Difference | | p-value | Cohen's d | | Power |
|---|---|---|---|---|---|---|---|---|
|  |  |  | Mean (SD) | 95%CI |  | Point | 95%CI |  |
| Number of particles | 21208.2 (7596.9) | 5.6 (1.7) | 21202.7 (7596.9) | 15768.2–26637.1 p < 0.01 | p < 0.01 | 2.791 | 1.368–4.188 | 1.00 |

Mean (SD) was shown.

Abbreviation: LFVS, Laminar airflow ventilation system.

conditions. With its inbuilt virus-removal mechanism (HEPA filters), this novel ventilation system creates a sterile vertical laminar flow and potentially inhibits cluster generation associated with airborne transmission in the gym.

### Exercise during the COVID-19 crisis

Little is known about respiratory droplets during physical exercise [27–29]. In this study, a sweat lactate sensor was used to examine in detail the relationship between the number of respiratory droplets from exhaled air and anaerobic threshold (Figs 3 and 4). As reported by Seki et al., non-invasive and continuous monitoring of sweat lactate values during exercise using a sweat lactate sensor has been shown to strongly correlate with ventilatory threshold assessed with exhaled gas analysis [11]. It was shown that the number of expiratory particulates increased in all cases when exercise was continued at 85% of the predicted maximum heart rate. In about half of the cases, the significant rise in aerosol occurred beyond the LT, suggesting that the aerosol elevation timing depends on the subject's exercise style (e.g., breathing technique, muscle usage, etc.). Conversely, none of the patients had an aerosol elevation before LT. A high ventilation/perfusion mismatch is observed due to differences in blood flow and ventilation at rest. After the start of exercise, a mild increase in minute ventilation (VE) was comparable to the oxygen uptake ($VO_2$) increase because of improved ventilation/perfusion mismatch [30]. After the LT, VE increased independently of $VO_2$ because of a significant increase in $CO_2$ consumption ($VCO_2$), and the increased respiratory rate led to respiratory muscle fatigue [31]. Consequently, increased aerosol levels could be observed during exercise above the LT. Continuous exercise at 85% of the predicted maximum heart rate led to the persistent elevation of the respiratory droplets, following a sustainable increase in the $VO_2$ because of fatigue and decreased exercise efficiency [32]. These findings suggested that aerobic exercise below the LT should be promoted even during the COVID-19 crisis to strengthen the immune system. Conversely, anaerobic exercise indoors should be carefully practiced with more robust infection-control measures.

### New lifestyle during the COVID-19 crisis

The promotion of hand washing, and social distancing is the most effective for COVID-19 prevention. However, interpersonal communication is essential to maintain mental and physical well-being despite the fears of coronavirus transmission. The abovementioned system is useful for safely improving physical activity during the COVID-19 crisis [33]. Furthermore, this system could be adapted to places where face-to-face interaction is required.

### Limitations

The present study has some limitations. First, as a single-center observational study, there may have been a selection bias. Second, our study did not allow for an examination of the possibility of LFVS in preventing the COVID-19 infection. A further randomized study with a large sample size and long-term follow-up for the prevention of COVID-19 infection is required to overcome these limitations.

## Conclusion

Exercise before the LT showed no increase in respiratory droplets, but anaerobic exercise revealed a significant increase. LFVS enabled a significant decrease in respiratory droplets during anaerobic exercise in healthy subjects, which may be useful for safely improving physical activity during the COVID-19 crisis.

## Supporting information

**S1 Fig. Particulates from an ultrasonic humidifier using the laminar airflow ventilation system (LFVS).** Particulates from an ultrasonic humidifier using the laminar airflow ventilation system (LFVS). The concentration of particulates ($>0.3$ μm) from an ultrasonic humidifier before and after the activation of the LFVS (n = 10). [##]$p<0.001$ compared with the off LFVS.
(PDF)

**S1 Movie. The performance of the vertical laminar flow ventilation system using artificial droplets.**
(AVI)

**S1 File. Data set.**
(XLSX)

## Acknowledgments

The authors thank T. Iida, T. Yamada, and Y. Mita (who manage the Science Laboratory, Tokyo, Japan) for their technical assistance. We are grateful to Editage for editing this manuscript.

## Author Contributions

**Conceptualization:** Yoshinori Katsumata, Eiji Kobayashi.

**Data curation:** Yoshinori Katsumata, Hiroki Okawara, Tomonori Sawada, Genki Ichihara.

**Formal analysis:** Yoshinori Katsumata, Hiroki Okawara, Tomonori Sawada, Genki Ichihara.

**Funding acquisition:** Yoshinori Katsumata, Eiji Kobayashi.

**Investigation:** Yoshinori Katsumata, Motoaki Sano, Eiji Kobayashi.

**Methodology:** Yoshinori Katsumata, Eiji Kobayashi.

**Project administration:** Yoshinori Katsumata.

**Resources:** Yoshinori Katsumata, Daisuke Nakashima.

**Software:** Daisuke Nakashima.

**Supervision:** Motoaki Sano, Keiichi Fukuda, Kazuki Sato.

**Validation:** Yoshinori Katsumata.

**Visualization:** Yoshinori Katsumata.

**Writing – original draft:** Yoshinori Katsumata.

**Writing – review & editing:** Motoaki Sano, Eiji Kobayashi.

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
