## [Decision Letter · Decision Letter 0]

7 Jul 2021

PONE-D-21-09706

Laminar flow ventilation system to prevent airborne infection during exercise in the COVID-19 crisis: a single-center observational study

PLOS ONE

Dear Dr. KATSUMATA,

Thank you for submitting your manuscript to PLOS ONE. After careful consideration, we feel that it has merit but does not fully meet PLOS ONE’s publication criteria as it currently stands. Therefore, we invite you to submit a revised version of the manuscript that addresses the points raised during the review process.

Please make corrections as suggested by reviewer.

We look forward to receiving your revised manuscript.

Kind regards,

Davor Plavec, MD, MSc, PhD, Prof.

Academic Editor

PLOS ONE

Journal Requirements:

2. In your Methods section, please provide additional information about the participant recruitment method and the demographic details of your participants. Please ensure you have provided sufficient details to replicate the analyses such as: a) the recruitment date range (month and year), b) a description of any inclusion/exclusion criteria that were applied to participant recruitment, c) a statement as to whether your sample can be considered representative of a larger population, d) a description of how participants were recruited."

3. During the internal evaluation of the manuscript we have noted in the Methods section that one patient experienced a subdural hematoma. Please provide further clarification regarding whether this was a direct result of the exercise intervention. Please provide details regarding any treatment which the participant received.

4. Please provide additional details regarding participant consent. In the ethics statement in the Methods and online submission information, please ensure that you have specified: 1) whether the ethics committee approved the verbal/oral consent procedure, 2) why written consent could not be obtained, and 3) how verbal/oral consent was recorded. If your study included minors, please state whether you obtained consent from parents or guardians in these cases. If the need for consent was waived by the ethics committee, please include this information.

5. Thank you for providing the following Funding Statement: 

"Yoshinori Katsumata has a financial relationship with Kimura Memorial Heart Foundation Research Grant for 2019, Suzuken Memorial Foundation, Foundation for Total Health Promotion, and Research Grant for Public Health Science. Eiji Kobayashi has a financial relationship with Nippon Medical & Chemical Instruments Co. Ltd. Motoaki Sano, Hiroki Okawara, Tomonori Sawada, Genki Ichihara, and Kazuki Sato declare that they have no competing interests."

We note that one or more of the authors is affiliated with the funding organization, indicating the funder may have had some role in the design, data collection, analysis or preparation of your manuscript for publication; in other words, the funder played an indirect role through the participation of the co-authors.

If the funding organization did not play a role in the study design, data collection and analysis, decision to publish, or preparation of the manuscript and only provided financial support in the form of authors' salaries and/or research materials, please review your statements relating to the author contributions, and ensure you have specifically and accurately indicated the role(s) that these authors had in your study in the Author Contributions section of the online submission form. Please make any necessary amendments directly within this section of the online submission form.  Please also update your Funding Statement to include the following statement: “The funder provided support in the form of salaries for authors [insert relevant initials], but did not have any additional role in the study design, data collection and analysis, decision to publish, or preparation of the manuscript. The specific roles of these authors are articulated in the ‘author contributions’ section.”

If the funding organization did have an additional role, please state and explain that role within your Funding Statement.

Please also provide an updated Competing Interests Statement declaring this commercial affiliation along with any other relevant declarations relating to employment, consultancy, patents, products in development, or marketed products, etc. 

7. Your ethics statement should only appear in the Methods section of your manuscript. If your ethics statement is written in any section besides the Methods, please delete it from any other section.

8. We note that Figure 1 in your submission contain copyrighted images. All PLOS content is published under the Creative Commons Attribution License (CC BY 4.0), which means that the manuscript, images, and Supporting Information files will be freely available online, and any third party is permitted to access, download, copy, distribute, and use these materials in any way, even commercially, with proper attribution. For more information, see our copyright guidelines: http://journals.plos.org/plosone/s/licenses-and-copyright.

We recommend that you contact the original copyright holder with the Content Permission Form (http://journals.plos.org/plosone/s/file?id=7c09/content-permission-form.pdf) and the following text: “I request permission for the open-access journal PLOS ONE to publish XXX under the Creative Commons Attribution License (CCAL) CC BY 4.0 (http://creativecommons.org/licenses/by/4.0/). Please be aware that this license allows unrestricted use and distribution, even commercially, by third parties. Please reply and provide explicit written permission to publish XXX under a CC BY license and complete the attached form.”

Please upload the completed Content Permission Form or other proof of granted permissions as an "Other" file with your submission. In the figure caption of the copyrighted figure, please include the following text: “Reprinted from [ref] under a CC BY license, with permission from [name of publisher], original copyright [original copyright year].”

Reviewers' comments:

Reviewer's Responses to Questions

**Comments to the Author**

1. Is the manuscript technically sound, and do the data support the conclusions?

Reviewer #1: Yes

2. Has the statistical analysis been performed appropriately and rigorously? 

Reviewer #1: Yes

3. Have the authors made all data underlying the findings in their manuscript fully available?

Reviewer #1: Yes

4. Is the manuscript presented in an intelligible fashion and written in standard English?

Reviewer #1: Yes

5. Review Comments to the Author

Reviewer #1: I think it is a very valuable paper especially in regard to present situation. Though I would be very happy if you could perform Power analysis (then you do not have to say in limitations that you have a small sample)

Also regarding the language. Do not use WE did this, WE did that. Please use impersonal, passive voice.

6. PLOS authors have the option to publish the peer review history of their article (what does this mean?). If published, this will include your full peer review and any attached files.

Reviewer #1: No

---

## [Author Response · Author response to Decision Letter 0]

2 Aug 2021

To Reviewer #1

#1. I think it is a very valuable paper especially in regard to present situation. Though I would be very happy if you could perform Power analysis (then you do not have to say in limitations that you have a small sample)

Response: 

Thank you very much for pointing this out. As suggested, a power analysis was performed. Based on a pre-performed Shapiro-wilk test, multiple comparisons of changes in the number of airborne particulates involving each incremental exercise period from the warm-up were made using the repeated analysis of variance with the Dunnet test as a post-hoc test. It revealed a significant and gradual increase in the respiratory droplets with time with a large effect size. Also, Cohen’s d was calculated using the value of t for paired t test to compare the droplet concentrations from spray or from the oral cavity during vigorous exercise before and after the activation of LFVS. We revised the related parts of the Materials and Methods, Results, Discussion, and Table, as shown below:

-Method section (Statistical analyses)-

The results are presented as means with standard deviations for continuous variables and as percentages for categorical variables, as appropriate. Based on a pre-performed Shapiro-wilk test, multiple comparisons of changes in the number of airborne particulates involving each incremental exercise period from the warm-up were made using the repeated analysis of variance with the Dunnet test as a post-hoc testSteel test. Student’s paired t-test was used to compare the droplet concentrations from spray or from the oral cavity during vigorous exercise before and after the activation of LFVS. We calculated Cohen’s d using the value of t for paired t test. SPSS, version 25.0 (SPSS Inc., Chicago, Illinois), was used for analysis, and p<0.05 (2-sided) was set to define statistical significance.

-Result section-

Monitoring particulates during exercise

Baseline characteristics of the subjects are summarized in Table 1. The subjects were predominantly male (80%), with an average age of 29 ± 12 years. Figure 3 shows the representative data for the particulates, lactate in sweat, and heart rate during exercise. A steady number of particulates before the LT was followed by a significant and gradual increase in the respiratory droplets after the LT, particularly during anaerobic exercise with a large effect size (Fig. 3 and Table 2). 

-Result section-

Effect of the LFVS on the particulates during vigorous exercise

The LFVS was activated during a constant load of exercise intensity above the LT, in which the concentration of particulates in the exhaled air increased. Notably, particulates exhaled during exercise were almost completely removed by the LFVS (>0.3 μm: 2120800±759700 vs. 560 ± 170; p<0.001, p<0.001, n=10; Fig. 4 and Table 3).

-Discussion section-

Limitations

The present study has some limitations. First, as a single-center observational study, there may have been a selection bias. Second, the number of subjects was small. Third, our study did not allow for an examination of the possibility of LFVS in preventing the COVID-19 infection. A further randomized study with a large sample size and long-term follow-up for the prevention of COVID-19 infection is required to overcome these limitations.

#2. Also regarding the language. Do not use WE did this, WE did that. Please use impersonal, passive voice.

Response:

As you mentioned, we revised the manuscript to remove the use of “We” when not appropriate. 

 

Journal Requirements:

Response: 

As suggested, we revised the manuscript in accordance with PLOS ONE's style requirements.

2. In your Methods section, please provide additional information about the participant recruitment method and the demographic details of your participants. Please ensure you have provided sufficient details to replicate the analyses such as: a) the recruitment date range (month and year), b) a description of any inclusion/exclusion criteria that were applied to participant recruitment, c) a statement as to whether your sample can be considered representative of a larger population, d) a description of how participants were recruited."

3. During the internal evaluation of the manuscript we have noted in the Methods section that one patient experienced a subdural hematoma. Please provide further clarification regarding whether this was a direct result of the exercise intervention. Please provide details regarding any treatment which the participant received.

Response: 

As suggested, we have revised the Methods section, as shown below:

-Method Section-

Subjects aged 20-80 years were recruited via a web system in October 2020. Exclusion criteria included receiving medication and having comorbidities, such as hypertension, diabetes, or active lung diseases. Twenty healthy subjects were enrolled recruited in this study The subjects had a broad spectrum of aerobic capacities and fitness levels, including athletes, and had no comorbidities, such as hypertension, diabetes, or active lung diseases. One of them had experienced a subdural hematoma 6 months ago. At the time he participated in this study, he was cured of his disease and was not receiving ongoing treatment. In addition, he was actively engaged in daily exercise. All of them received no medication. They can be considered representative of a larger population. The study protocol was approved by the Institutional Review Board (IRB) of Keio University School of Medicine [permission number; 20190229], and the study was conducted in accordance with the Declaration of Helsinki. Subjects provided verbal informed consent, because the IRB approved use of oral consent in accordance with Japanese guidance for clinical research. Verbal consents were recorded as experimental notes in this study.

4. Please provide additional details regarding participant consent. In the ethics statement in the Methods and online submission information, please ensure that you have specified: 1) whether the ethics committee approved the verbal/oral consent procedure, 2) why written consent could not be obtained, and 3) how verbal/oral consent was recorded. If your study included minors, please state whether you obtained consent from parents or guardians in these cases. If the need for consent was waived by the ethics committee, please include this information.

Response: 

Thank you very much for pointing this out. The IRB approved the use of oral consent in accordance with Japanese guidance for clinical research. The verbal consent was recorded as an experimental note in this study. Subjects aged 20-80 years old were recruited. We revised the Methods section, as shown below:

-Method Section-

Subjects aged 20-80 years were recruited via a web system in October 2020. Exclusion criteria included receiving medication and having comorbidities, such as hypertension, diabetes, or active lung diseases. Twenty healthy subjects were enrolled recruited in this study The subjects had a broad spectrum of aerobic capacities and fitness levels, including athletes, and had no comorbidities, such as hypertension, diabetes, or active lung diseases. One of them had experienced a subdural hematoma 6 months ago. At the time he participated in this study, he was cured of his disease and was not receiving ongoing treatment. In addition, he was actively engaged in daily exercise. All of them received no medication. They can be considered representative of a larger population. The study protocol was approved by the Institutional Review Board (IRB) of Keio University School of Medicine [permission number; 20190229], and the study was conducted in accordance with the Declaration of Helsinki. Subjects provided verbal informed consent because the IRB approved use of oral consent in accordance with Japanese guidance for clinical research. Verbal consents were recorded as experimental notes in this study.

5. Thank you for providing the following Funding Statement: 

"Yoshinori Katsumata has a financial relationship with Kimura Memorial Heart Foundation Research Grant for 2019, Suzuken Memorial Foundation, Foundation for Total Health Promotion, and Research Grant for Public Health Science. Eiji Kobayashi has a financial relationship with Nippon Medical & Chemical Instruments Co. Ltd. Motoaki Sano, Hiroki Okawara, Tomonori Sawada, Genki Ichihara, and Kazuki Sato declare that they have no competing interests."

We note that one or more of the authors is affiliated with the funding organization, indicating the funder may have had some role in the design, data collection, analysis or preparation of your manuscript for publication; in other words, the funder played an indirect role through the participation of the co-authors.

If the funding organization did not play a role in the study design, data collection and analysis, decision to publish, or preparation of the manuscript and only provided financial support in the form of authors' salaries and/or research materials, please review your statements relating to the author contributions, and ensure you have specifically and accurately indicated the role(s) that these authors had in your study in the Author Contributions section of the online submission form. Please make any necessary amendments directly within this section of the online submission form. Please also update your Funding Statement to include the following statement: “The funder provided support in the form of salaries for authors [insert relevant initials], but did not have any additional role in the study design, data collection and analysis, decision to publish, or preparation of the manuscript. The specific roles of these authors are articulated in the ‘author contributions’ section.”

If the funding organization did have an additional role, please state and explain that role within your Funding Statement.

Please also provide an updated Competing Interests Statement declaring this commercial affiliation along with any other relevant declarations relating to employment, consultancy, patents, products in development, or marketed products, etc. 

Response: 

As suggested, we revised the Methods section, as shown below:

Funding Statement

The laminar airflow ventilation system was developed in collaboration with E.K. and Nippon Medical & Chemical Instruments Co. Ltd. This work was partly supported by a Grant-in-Aid from Scientific Research from the Japan Agency for Medical Research and Development [ID. 21ek0210130h0003] and by Kimura Memorial Heart Foundation Research Grant for 2019 [N/A], Suzuken Memorial Foundation [N/A], Foundation for Total Health Promotion [N/A], and Research Grant for Public Health Science [N/A].

The funders provided support in the form of financial supports for authors [Y.K., E.K.], but did not have any additional role in the study design, data collection and analysis, decision to publish, or preparation of the manuscript. The specific roles of these authors are articulated in the ‘author contributions’ section.

Competing Interests Statement

Yoshinori Katsumata has a financial relationship with Kimura Memorial Heart Foundation Research Grant for 2019, Suzuken Memorial Foundation, Foundation for Total Health Promotion, and Research Grant for Public Health Science. Eiji Kobayashi has a financial relationship with Nippon Medical & Chemical Instruments Co. Ltd. Motoaki Sano, Hiroki Okawara, Tomonori Sawada, Genki Ichihara, and Kazuki Sato declare that they have no competing interests.

This does not alter our adherence to PLOS ONE policies on sharing data and materials.

Response: 

As suggested, we uploaded the minimal data set. And, we revised the manuscriot, as shown below:

Availability of data and material

The datasets are within the manuscript and its Supporting Information files. 

The datasets generated during and/or analyzed during the current study are not publicly available but are available from the corresponding author on reasonable request. 

7. Your ethics statement should only appear in the Methods section of your manuscript. If your ethics statement is written in any section besides the Methods, please delete it from any other section.

Response:

As suggested, we revised the manuscript.

8. We note that Figure 1 in your submission contain copyrighted images. All PLOS content is published under the Creative Commons Attribution License (CC BY 4.0), which means that the manuscript, images, and Supporting Information files will be freely available online, and any third party is permitted to access, download, copy, distribute, and use these materials in any way, even commercially, with proper attribution. For more information, see our copyright guidelines: http://journals.plos.org/plosone/s/licenses-and-copyright.

We recommend that you contact the original copyright holder with the Content Permission Form (http://journals.plos.org/plosone/s/file?id=7c09/content-permission-form.pdf) and the following text: “I request permission for the open-access journal PLOS ONE to publish XXX under the Creative Commons Attribution License (CCAL) CC BY 4.0 (http://creativecommons.org/licenses/by/4.0/). Please be aware that this license allows unrestricted use and distribution, even commercially, by third parties. Please reply and provide explicit written permission to publish XXX under a CC BY license and complete the attached form.”

Please upload the completed Content Permission Form or other proof of granted permissions as an "Other" file with your submission. In the figure caption of the copyrighted figure, please include the following text: “Reprinted from [ref] under a CC BY license, with permission from [name of publisher], original copyright [original copyright year].”

Response:

As suggested, we completed the Content Permission Form.

Response:

Thank you. We have not cited papers that have been retracted. No changes were made to the references.

---

## [Decision Letter · Decision Letter 1]

6 Sep 2021

Laminar flow ventilation system to prevent airborne infection during exercise in the COVID-19 crisis: a single-center observational study

PONE-D-21-09706R1

Dear Dr. Katsumata,

We’re pleased to inform you that your manuscript has been judged scientifically suitable for publication and will be formally accepted for publication once it meets all outstanding technical requirements.

Kind regards,

Davor Plavec, MD, MSc, PhD, Prof.

Academic Editor

PLOS ONE

Additional Editor Comments (optional):

After making suggested corrections the manuscript is now acceptable for publication.

Reviewers' comments:

Reviewer's Responses to Questions

**Comments to the Author**

1. If the authors have adequately addressed your comments raised in a previous round of review and you feel that this manuscript is now acceptable for publication, you may indicate that here to bypass the “Comments to the Author” section, enter your conflict of interest statement in the “Confidential to Editor” section, and submit your "Accept" recommendation.

Reviewer #1: All comments have been addressed

2. Is the manuscript technically sound, and do the data support the conclusions?

Reviewer #1: Yes

3. Has the statistical analysis been performed appropriately and rigorously? 

Reviewer #1: Yes

4. Have the authors made all data underlying the findings in their manuscript fully available?

Reviewer #1: Yes

5. Is the manuscript presented in an intelligible fashion and written in standard English?

Reviewer #1: Yes

6. Review Comments to the Author

Reviewer #1: Thank you for addressing my remarks regarding impersonal language, power analysis and limitations

.

7. PLOS authors have the option to publish the peer review history of their article (what does this mean?). If published, this will include your full peer review and any attached files.

Reviewer #1: No

---

## [Editor Report · Acceptance letter]

19 Oct 2021

PONE-D-21-09706R1 

Laminar flow ventilation system to prevent airborne infection during exercise in the COVID-19 crisis: a single-center observational study 

Dear Dr. Katsumata:

I'm pleased to inform you that your manuscript has been deemed suitable for publication in PLOS ONE. Congratulations! Your manuscript is now with our production department. 

Kind regards, 

on behalf of

Dr. Davor Plavec 

Academic Editor

PLOS ONE